# The Relationships between Prospection, Self-Efficacy, and Depression in College Students with Cross-Lagged Analysis

**DOI:** 10.3390/ijerph192214685

**Published:** 2022-11-09

**Authors:** Zhuo-Ya Yang, Ya-Ting Wang, Lei Xia, Ying-Can Zheng, Zheng-Zhi Feng

**Affiliations:** 1Department of Basic Psychology, School of Medical Psychology, Army Medical University, Chongqing 400038, China; 2School of Medical Psychology, Army Medical University, Chongqing 400038, China; 3Business School, National University of Singapore, Singapore 119245, Singapore; 4Department of Developmental Psychology for Armyman, School of Medical Psychology, Army Medical University, Chongqing 400038, China

**Keywords:** depression, prospection, self-efficacy, college students, longitudinal study

## Abstract

Depression is one of the most prevalent and disabling mental health problems in college students. Previous studies have established cross-sectional associations between negative bias in prospection e.g., increased negativity in future simulation, low self-efficacy, and depressive symptoms. Nevertheless, the temporal bidirectional associations between them are rarely examined. In the current study, we collected valid data on 276 college students at two time points within a 10 week interval. Cross-lagged panel analysis was applied to investigate the relationships between proportions of negative future events, levels of self-efficacy, and depressive symptoms. Results suggested depressive symptoms predict subsequent proportions of negative prospections and levels of self-efficacy. Inversely, neither prospection nor self-efficacy predicted depression. Temporal correlations between prospection and self-efficacy were also not significant. Since this is one of the first studies that attempts to figure out temporal links between these mutually informing factors, more longitudinal research is needed to draw a firm conclusion. This study provides new insights into the relationship between negative biases in cognitions and depressive symptoms and highlights the need to intervene early with depressive symptoms before any possible cognitive distortions in college students.

## 1. Introduction

Depression is one of the most prevalent and disabling mental health problems in college students [1]. In China, the prevalence of depression among college students is found to be over 20 percent and has increased over the past few years [1,2]. Depressive symptoms can impair cognitive and emotional functioning [3], and results in both short-term and long-term adverse consequences, including poor academic and occupational attainment [4], low life quality [5], and suicide [6] in college students. Therefore, it is of great importance to investigate the relationships of the possible psychological factors driving and maintaining depression in college students in order to facilitate prevention and early intervention and improve their mental health.

A negative view of the future, of the self, and of the world, are not the only hallmark symptoms of depression, but also causal factors contributing to depression, as Beck’s (1974) negative cognitive triad model assumed [7]. Negative bias in prospection (e.g., increased negativity in generating future episodes) and low self-efficacy (lack of belief in your own abilities to perform the actions required to achieve goals) have been consistently found in both clinical and subclinical depression [8,9,10,11]. Evidence also supports that prospection and self-efficacy could be associated with each other [12,13]. However, previous research mainly focused on the cross-sectional relationships between any two of the three mutually informing factors (e.g., depression and prospection, depression and self-efficacy, or prospection and self-efficacy). The longitudinal bidirectional associations between prospection, self-efficacy, and depression have not been fully explored. 

### 1.1. Prospection and Depression

A close association between negative bias in prospection and depression has been established in cross-sessional studies. Negative bias in prospection has proved to be present in both clinical and subclinical depression; for future events construction, excessive negative future episodes have been found [8,14]; additionally, for future events elaboration, vivid negative images, and elevated negative emotions have been suggested by previous studies [8,15]. As for the longitudinal studies, previous research has demonstrated that hopelessness can significantly predict depressive symptoms after both a short interval, five weeks in undergraduates [16] and a two or four year period in high-risk adolescence [17]. On the other hand, depression can predict negative appraisals about personal projects (e.g., uncapable of accomplishing projects two years later) in a nonclinical sample [18]. These studies supported the “Depression and Prospection” model, which proposed that negative biases in simulations, expectations, and beliefs about the future drive depression, and in turn, depression maintains these deficits, to form a vicious cycle [9]. However, previous research mainly focused on the longitudinal relationship between general expectations or beliefs e.g., hopeless and depression. There is a paucity of research regarding temporal associations between negative bias in future simulation and depression. 

Gender differences have been found in prospection and depressive symptoms experience. Previous research suggests women report more future events of personal importance than men in both dysphoric and non-dysphoric people [8]. Moreover, women tend to report more depressive symptoms than men [19]. However, it has been rarely tested whether gender has an impact on the temporal association between prospection and depressive symptoms.

### 1.2. Self-Efficacy and Depression

Substantial evidence has proved that self-efficacy is concurrently correlated with depression [20,21], while their causality relationship is less established in longitudinal studies. The social cognitive theory suggests a bidirectional link between self-efficacy and depressive symptoms [22]. It argues that a low level of self-efficacy might result in depressive feelings through a discrepancy in expectations and perceived abilities; in turn, depressive feelings could also impede self-efficacy [22]. However, not all previous findings fully support this proposition. Several studies have indicated self-efficacy could predict depressive symptoms [23,24], while others failed to find the prospective association of self-efficacy to depression [11,25]. 

In general, women tend to have lower levels of self-efficacy when compared to men [20,24,26]. Moreover, self-efficacy was negatively correlated with depressive symptoms for women, but not for men in a cross-sectional design [27]. A longitudinal study has implied no gender difference in cross-lagged associations between self-efficacy and depression levels in healthy adolescence [25]. More efforts should be exerted to figure out gender differences in the link between self-efficacy and depression in both nonclinical and clinical samples for precise interventions.

### 1.3. Prospection and Self-Efficacy

With regard to the relationship between prospection and self-efficacy, self-efficacy might be maintained partially through memory retrieval and future simulation that support the current self [28]. On the other hand, according to the self-memory-system theory, self-perception could serve as a crucial guide in the construction of past and future episodes [29]. Previous studies have demonstrated that boosting self-efficacy improves specificity and positivity towards future events in undergraduate students [12] and individuals with PTSD [13]. However, little is known about the temporal relationship between prospection and self-efficacy. It also remains unclear how prospection and self-efficacy interact in the context of depression. McMichael et al. (2021) suggested the cross-sectional relationships between the vividness of future events and self-efficacy might be the same for both sexes. Since it has rarely been explored previously, further research should be conducted to address gender effects [30].

### 1.4. The Current Study

The current study investigated the longitudinal and bidirectional correlations between prospection, self-efficacy, and depressive symptoms in college students using cross-lagged panel analysis. We adopted a short time interval (10 weeks) to better capture the possible links between fluctuations of depressive symptoms and other variables, as several studies have suggested [18,25]. We hypothesized that: (1) negative biases in prospection (e.g., excessive negative events) would predict depression severity, assessed ten weeks later, or vice versa; (2) self-efficacy and depression would also prospectively predict each other across a ten week period; (3) prospection and self-efficacy would be positively correlated at both time points. Since there is a lack of longitudinal evidence, no specific hypothesis was formed on the temporal association between prospection and self-efficacy. In addition, it was examined whether gender affected above associations. This study will provide new insights into the temporal links between negative biases in cognitions and depressive symptoms.

## 2. Materials and Methods

### 2.1. Participants

College students were recruited in Chongqing, China. Participants responded to on-line links on the Wenjuanxing platform (https://www.wjx.cn/, accessed on 14 September 2021), providing informed consent and answered the self-report questionnaires. At Time 1 (T1), a total of 388 participants completed all measurements. The exclusion criteria were as follows: (1) score on either lie detection item was greater than 4 (N = 14); (2) completion time was shorter than 157.65 s (2.5th percentile) (N = 11) [31]; (3) the response was “No” to the last question “Did you read each item carefully and answer it honestly?” (N = 9); (4) responses to the SCEFT were repetitive or omitted for more than half of items (N = 19). After this, 335 valid participants remained who were invited to complete the same on-line survey again 10 weeks later at Time 2 (T2). More than 90% participants (N = 321) responded at T2, and the above exclusion criteria were also applied to the T2 data. The final sample comprised 276 participants (age: M = 19.01, SD = 1.46, 153 females) who had valid data at both T1 and T2. Ethical approval for the study was granted by the Ethics Committee of Army Medical University, Chongqing, China.

### 2.2. Measurements

#### 2.2.1. The Depression, Anxiety, and Stress Scale-21(DASS-21)

The DASS-21 is a self-report scale assessing levels of depression, anxiety, and stress [32]. The Chinese version of DASS-21 was applied in the current study [32]. The DASS-21 consists of three subscales of depression, anxiety, and stress, each of which contains seven items (depression: item 3, 5, 10, 13, 16, 17, 21; anxiety: item 2, 4, 7, 9, 15, 19, 20; and stress: item 1, 6, 8, 11, 12, 14, 18). Items are scored from 0 (“did not apply to me at all”) to 3 (“applied to me very much or most of the time”), with higher scores indicating more symptoms. The Chinese version of the DASS-21 is shown to possess a good reliability and validity [2,33]. In the present study, the depression subscale (DASS-D) was applied to measure depressive symptoms at both T1 (Cronbach’s α = 0.85) and T2 (Cronbach’s α = 0.84).

#### 2.2.2. The Sentence Completion for Events in the Future Test (SCEFT)

The SCEFT [34] includes 11 sentence stems for participants to generate possible future events. For example, “In the future I will…”. Participants were instructed to complete the sentence in any form, with each sentence reflecting contents on different topics. Responses were rated for specificity and emotional valence as Chen et al. (2016)’s study [34]. Specificity was coded into five categories: (1) a specific event happening at a particular time and place; (2) an extended event lasting for more than one day; (3) a categorical event reflecting general contents in a category; (4) semantic associates; and (5) omission. The emotional valence was coded into positive, negative, and neutral. The Chinese version of SCEFT used in this study was demonstrated to have good psychometric properties [35,36].

#### 2.2.3. The General Self-Efficacy Scale (GSES)

The GSES is a self-report instrument comprising 10 items [37]. It is scored on a 4 point Likert-type scale ranging from 1 (not at all true) to 4 (exactly true). The total scores ranged from 10 to 40, with higher scores indicating a greater self-efficacy. The Chinese version of the GSES [38] is validated in both general populations [39] and in clinical settings [40]. In this study, the internal consistency was good for the GSES at both T1 (Cronbach’s α = 0.94) and T2 (Cronbach’s α = 0.95).

### 2.3. Data Analysis

Data analyses were performed using SPSS version 26.0 (SPSS Inc., Chicago, IL, USA) and Mplus 7.4 [41]. Two recruited raters interdependently coded responses for the SCEFT according to the scoring manual for specificity and emotion valence. The two recruiters were both blind to the study’s design. Participants’ performance was mainly measured by the proportions of events generated in each specificity category and each valence type. The inter-rater reliabilities were good (Cohen’s Kappa, specificity: 0.85; emotional valence: 0.88) [42]. Discrepancies were reconciled through discussion or consulting with a third party until reaching full consensus. First, descriptive, and correlational analyses were conducted relating to the main variables. Then cross-lagged panel models focusing on depressive symptoms, proportions of negative future events, and self-efficacy were further estimated. Additionally, gender as a moderator was added to the cross-lagged panel models.

## 3. Results

### 3.1. Descriptive and Correlational Findings

We found that the pattern for proportions of events in each specific category or emotional valence (e.g., for T1, specific: M = 0.23, SD = 0.14, extended: M = 0.27, SD = 0.15, categoric: M = 0.22, SD = 0.15, semantic: M = 0.25, SD = 0.16; positive: M = 0.50, SD = 0.20, negative: M = 0.03, SD = 0.06, neutral: M = 0.44, SD = 0.20) was similar to findings in nonclinical groups in a previous studies [35,36], which further proved the effectiveness of our study manipulation. The SCEFT performance at both time points are present in Appendix A. We did not find significant correlations between specificity and depression. In order to investigate associations between negative bias in prospection, self-efficacy, and depressive symptoms, we focused on proportions of negative future events, the GSES scores, and the DASS-D scores in the following analysis. Table 1 provides descriptive statistics and correlations among these main variables. The finding that proportions of negative future events were concurrently related to self-efficacy at both time points (T1: r = 0.15, *p* = 0.01; T2: r = 0.25, *p* < 0.001), supported the hypothesis 3.

### 3.2. Measure Invariance

The measurement invariance was examined, which is a precondition to adequately test cross-lagged effects [43,44]. Results of tests for configural and metric invariance are provided in Table 2. Models specifying the same factor structure across time (configural invariance) all demonstrated an acceptable to good fit to the data. Furthermore, setting the factor loadings equal across time (metric invariance) did not significantly change the model fit for self-efficacy, as demonstrated by the nonsignificant change in χ^2^ (Δχ^2^ (9) = 11.66, *p* = 0.23). Although there was a significant change in χ^2^ for depression (Δχ^2^ (6) = 13.95, *p* = 0.03), the modification indices for the metric model are still ideal. Therefore, our dataset generally evidences its measurement invariance across time.

### 3.3. Cross-Lagged Models

Table 3 provides the results of the cross-lagged panel models. Because of the potential for multicollinearity between proportions of negative future events and self-efficacy to bias results, we specified both separate bivariate cross-lagged panel models (Model 1 and Model 2 for proportions of negative future events and self-efficacy, respectively) and a cross-lagged panel model that included both proportions of negative future events and self-efficacy in one analysis (Model 3). Each of these models includes three types of parameter estimates: the stabilities of each latent construct across time, the cross-lagged effects, and predictor correlations at Time 1. Figure 1 provides results from Model 3, showing all of these described elements. All three models tested provided a good fit to the data and the results from the overall cross-lagged model (Model 3) closely mirror those from the separate bivariate cross-lagged models (Model 1 and Model 2; see Table 3).

We focus on the results of Model 3 in evaluating hypothesis 1. As shown in Model 1 and Model 3 of Table 3 and Figure 1, depression at T1 was positively related to proportions of negative future events at T2 (Model 1: β = 0.01, *p* < 0.001; Model 3: β = 0.01, *p* < 0.001), but T1 proportions of negative future events was not related to depression at T2 (Model 1: β = 3.63, *p* = 0.17; Model 3: β = 3.62, *p* = 0.17). This pattern of findings supports that the direction seems to occur from depression to proportions of negative future events, rather than vice versa.

As also shown in Model 2 and Model 3 of Table 3 and Figure 1, depression at T1 was positively related to self-efficacy at T2 (Model 2: β = −0.20, *p* < 0.001; Model 3: β = −0.18, *p* = 0.008), but T1 self-efficacy was not related to depression at T2 (Model 2: β = −0.00, *p* = 0.90; Model 3: β = −0.00, *p* = 0.95). This pattern of findings supports that the direction seems to occur from depression to self-efficacy, rather than from self-efficacy to depression, which is partially consistent with hypothesis 2.

### 3.4. Moderating Effect of Gender

As shown in Model 1 and Model 3 of Table 4, we found that the cross-lagged effect of depression on proportions of negative future events was stronger (Model 1: difference = 0.01, *p* < 0.001; Model 3: difference = 0.01, *p* = 0.07) for female (Model 1: β = 0.01, *p* < 0.001; Model 3: β = 0.01, *p* < 0.001) than male (Model 1: β = 0.004, *p* = 0.01; Model 3: β = 0.004, *p* = 0.02). However, the cross-lagged effect of proportions of negative future events on depression was not affected by gender (Model 1: difference = −4.77, *p* = 0.32; Model 3: difference = −5.57, *p* = 0.25).

## 4. Discussion

The current study investigated the reciprocal effects of prospection, self-efficacy and depression in college students with a longitudinal design including two time points over 10 weeks. Results indicated that high levels of depressive symptoms predicted high proportions of negative future events, while negative biases in prospection did not predict depression. Moreover, depression severity negatively predicted self-efficacy, but not inversely. We also found that although prospection and self-efficacy were positively correlated at each time point, their cross-lagged associations were not significant. Additionally, results suggested a stronger prediction effect of depression to proportions of negative future events in women than in men. Several implications can be provided according to these findings: (1) significant prediction effects of depression on cognitive distortions in the nonclinical sample over a short interval may highlight the necessity and importance of early interventions; (2) gender-specific interventions should be adopted to better deal with negative future and self in depression.

### 4.1. Prospection and Depression

Our findings partially supported hypothesis 1, suggesting a negative bias in prospection might be the consequence of depression. It is in line with the mood-congruent cognitions theory, which assumes that depressive mood provides the basis for depressive cognitions rather than vice versa [45]. Furthermore, depression might influence prospection through other ways. For instance, depression could create stressful experiences and interpersonal conflicts [46]. Therefore, those negative experiences might provide more raw materials for excessive negative future simulations since future events are constructed from the recombination of past elements [47]. This assumption is supported by our results demonstrating a significant cross-lagged effect of depression on proportions of negative future events.

Contrary to hypothesis 1, we did not find any significant effect in proportions of negative future events on depressive symptoms over an interval of 10 weeks. It might partially be on account of methodological issues. First, the cross-lagged model included the concurrent correlations and stability in prospection, depression, and self-efficacy domains over time, which provided a more detailed picture of the potentially complex associations than previous studies. As the present study is one of the first studies examining the bidirectional temporal relationships between these three factors, more longitudinal research is needed to draw a firm conclusion. Second, samples and durations of the intervals applied could exert an important influence on cross-lagged relationships. For example, the stable level of depressive symptoms across a short interval in our nonclinical samples might reduce the power of prospection to explain its change [18].

### 4.2. Self-Efficacy and Depression

Our results also partially sustained hypothesis 2 and the social cognitive theory, indicating that depressive symptoms negatively predict self-efficacy. According to the social cognitive theory, depression is usually associated with impaired functioning in life domains (e.g., work and social interactions), which could lead to increased negative experiences [22]. As a result, self-efficacy might decrease in depressed people. In addition, the mood-congruent cognitions theory [45] might also be another possible explanation.

The finding that the effect of self-efficacy on subsequent depressive symptoms was not significant is inconsistent with our assumptions. Both the social cognitive theory [44] and the cognitive theory of depression [48] postulate an etiologic role of negative self-perception in depression, including an interacting factor, i.e., prospection, in the cross-lagged model, which might have an impact on the results. However, a previous study examining the cross-lagged association between self-efficacy and depression also suggested that self-efficacy did not predict depressive symptoms in healthy adolescents [25]. Other methodological differences (e.g., samples and time intervals) discussed above might be possible explanations as well. Nevertheless, some have argued that low self-efficacy could be a symptom of, rather than a contributor to depression [49]. This viewpoint is supported by studies which found self-efficacy enhanced as depressive symptoms remitted in clinical depressed individuals [50,51].

### 4.3. Prospection and Self-Efficacy

Hypothesis 3 was proven by the finding that a negative bias in prospection and low self-efficacy are concurrently correlated at both T1 and T2, which is consistent with the previous studies [12,13]. Interestingly, we did not find significant temporal associations between prospection and self-efficacy. Since it is rarely examined in the literature, further longitudinal research adopting different samples, time intervals, and measurements are needed to address this issue. It is noteworthy that high proportions of negative future events and a lack of self-efficacy could be independently predicted by a baseline level of depressive symptoms. This might suggest that a negative view of future and of self are interactive but independent factors associated with depression, as Beck’s model suggested [7].

Our results also suggested an interesting gender difference, that is, although depression could predict proportions of negative future events in both women and men, the effect was stronger in women than in men. Previous studies suggest that different profiles of symptoms might present in depressed men and women, with women experiencing more sadness [52] and less pleasure [53] than men. Therefore, these different depressive presentations can have a different impact on both the subsequent cognitive and emotional functioning in women and men. Our findings might indicate depressive symptoms could exert more influence on a negative view of future in women than in men, which provides a reason for gender-specific interventions on depression.

### 4.4. Limitations and Future Directions

There are several limitations in the current study that we should acknowledge. First, since a nonclinical sample with a relatively low and stable level of depressive symptoms was adopted, subtle changes or small effect sizes might not be detected. Previous studies have implied cognitive impairments could predict depression only for individuals with high levels of depressive symptoms [54]. Therefore, diverse samples including healthy, subclinical, and clinical samples should be applied in future research. Second, our study included only two timepoints with a short interval. More longitudinal studies using different timepoints and intervals are needed to figure out these complex temporal associations. Finally, only self-report instruments were used in this study. Further investigations should include both subjective and objective measures to provide a more complete picture.

## 5. Conclusions

To conclude, this is, to our knowledge, one of the very first studies to examine longitudinal and bidirectional associations between depression, prospection, and self-efficacy in healthy undergraduates, which presents important preventative implications. We found that depressive symptoms could predict negative bias in prospection and low self-efficacy in a nonclinical sample, which address the importance of early intervention on depressive symptoms. Inversely, proportions of negative future events and levels of self-efficacy did not predict depressive symptoms in our study. More longitudinal research is needed that can test these bidirectional associations in different samples before firm conclusions are drawn.

## Figures and Tables

**Figure 1 ijerph-19-14685-f001:**
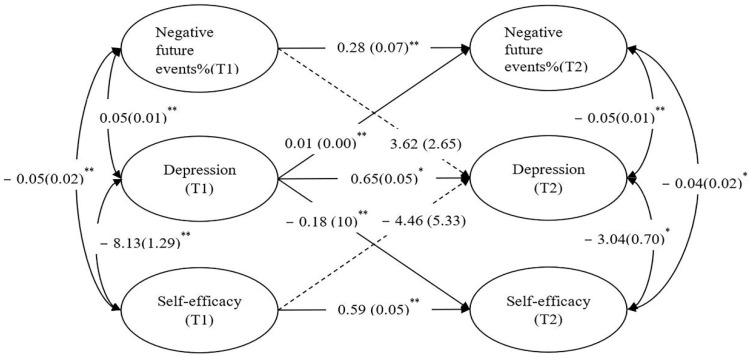
Cross-lagged model results for depression, proportions of negative future events and self-efficacy. Note. Values shown outside parentheses are unstandardized parameter estimates. Values shown within parentheses are standard error estimates. T1 = Time 1, T2 = Time 2. * *p* < 0.05. ** *p* < 0.01.

**Table 1 ijerph-19-14685-t001:** Means, SDs, and Correlations Among Study Variables.

	Mean	SD	1	2	3	4	5	6	7	8
1. Gender	0.55	0.50	1							
2. Age	19.01	1.46	−0.19 **	1						
3. Depression (T1)	9.30	3.10	−0.06	0.15 *	1					
4. Negative future events% (T1)	0.03	0.06	0.05	0.13 *	0.26 **	1				
5. Self-efficacy (T1)	26.29	6.35	−0.16 **	0.02	−0.41 **	−0.15 *	1			
6. Depression (T2)	9.59	3.14	−0.20 **	0.001	0.66 **	0.23 **	−0.28 **	1		
7. Negative future events% (T2)	0.04	0.07	0.02	0.05	0.36 **	0.31 **	−0.19 **	0.47 **	1	
8. Self-efficacy (T2)	26.65	6.24	−0.07	0.02	−0.35 **	−0.152 *	0.64 **	−0.39 **	−0.25 **	1

For Gender, 0 = male, 1 = female; negative future events%, proportions of negative future events in the SCEFT; * *p* < 0.05. ** *p* < 0.01.

**Table 2 ijerph-19-14685-t002:** Measurement Invariance Results.

	χ^2^	df	CFI	TLI	SRMR	Δdf	Δχ^2^
Depression							
Model 1: Configural invariance	78.39	28	0.97	0.95	0.03		
Model 2: Metric invariance	92.34	34	0.96	0.95	0.05	6	13.95 *
Self-efficacy							
Model 1: Configural invariance	290.40	70	0.95	0.93	0.04		
Model 2: Metric invariance	302.06	79	0.95	0.94	0.05	9	11.66

* *p* < 0.05.

**Table 3 ijerph-19-14685-t003:** Summary of Cross-Lagged Model Estimates.

Estimate	Model 1		Model 2		Model 3	
	Β (*p*)	se	Β (*p*)	se	Β (*p*)	se
** *Stabilities* **						
Depression	0.65 (0.00)	0.05	0.66 (0.00)	0.05	0.65 (0.00)	0.05
Negative future events%	0.28 (0.00)	0.07			0.28 (0.00)	0.07
Self-efficacy			0.59 (0.00)	0.05	0.59 (0.00)	0.05
** *Cross-lagged effects of depression* **						
Depression→Negative future events%	0.01 (0.00)	0.00			0.01 (0.00)	0.00
Depression→Self-efficacy			−0.20 (0.00)	0.05	−0.18 (0.08)	0.10
** *Cross-lagged effects of Negative future events%* **						
Negative future events%→Depression	3.63 (0.17)	2.65			3.62 (0.17)	2.65
Negative future events%→Self-efficacy					−4.46 (0.40)	5.33
** *Cross-lagged effects of self-efficacy* **						
Self-efficacy→Depression			−0.00 (0.90)	0.03	−0.000 (0.95)	0.03
Self-efficacy→Negative future events%					0.00 (0.54)	0.00
** *Predictor correlations at Time 1* **						
Depression↔Negative future events%	0.05 (0.00)	0.01			0.05 (0.00)	0.01
Depression↔Self-efficacy			−8.13 (0.00)	1.28	−8.13 (0.00)	1.29
Self-efficacy↔Negative future events%					−0.05 (0.01)	0.02
** *Disturbance correlations at Time 2* **						
Depression↔Negative future events%	0.05 (0.00)	0.01			0.05 (0.00)	0.01
Depression↔Self-efficacy			3.08 (0.00)	0.70	−3.04 (0.03)	0.70
Self-efficacy↔Negative future events%					−0.04 (0.03)	0.02

Values shown are unstandardized parameter estimates. Model 1 includes proportions of negative future events. Model 2 includes self-efficacy. Model 3 includes both proportions of negative future events and self-efficacy.

**Table 4 ijerph-19-14685-t004:** Moderating Effect of Gender.

Estimate	Model 1		Model 2		Model 3	
	Β (*p*)	se	Β (*p*)	se	Β (*p*)	se
** *Cross-lagged effects of depression* **						
Depression→Negative future events%						
Female	0.01 (0.00)	0.00			0.01 (0.00)	0.00
Male	0.00 (0.01)	0.00			0.004 (0.02)	0.00
Difference	0.01 (0.03)	0.00			0.01 (0.07)	0.00
Depression→Self-efficacy						
Female			−0.13 (0.34)	0.14	−0.08 (0.59)	0.14
Male			−0.25 (0.06)	0.13	−0.27 (0.05)	0.14
Difference			0.11 (0.53)	0.18	0.19 (0.32)	0.19
** *Cross-lagged effects of Negative future events%* **						
Negative future events%→Depression						
Female	2.07 (0.54)	3.39			1.42 (0.68)	3.42
Male	6.84 (0.06)	3.66			6.99 (0.06)	3.66
Difference	−4.77 (0.32)	4.78			−5.57 (0.25)	4.82
Negative future events%→Self-efficacy						
Female					−0.00 (0.44)	0.00
Male					0.00 (0.96)	0.00
Difference					−0.00 (0.58)	0.00
** *Cross-lagged effects of self-efficacy* **						
Self-efficacy→Depression						
Female			−0.04 (0.18)	0.03	−0.04(0.17)	0.03
Male			0.00 (0.90)	0.03	0.00(0.90)	0.04
Difference			−0.05 (0.28)	0.04	−0.05(0.28)	0.04
Self-efficacy→Negative future events%						
Female					−10.26 (0.16)	7.28
Male					1.71 (0.82)	7.84
Difference					−11.97 (0.27)	10.75

Values shown are unstandardized parameter estimates. Model 1 includes proportions of negative future events. Model 2 includes self-efficacy. Model 3 includes both proportions of negative future events and self-efficacy.

## Data Availability

The data presented in this study are available on request from the corresponding author. The data are not publicly available due to privacy restrictions.

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
