# Peer review of "The Relationships between Prospection, Self-Efficacy, and Depression in College Students with Cross-Lagged Analysis"

_ijerph, 2022, doi:10.3390/ijerph192214685_

Round 1
Reviewer 1 Report
I have reviewed your work from start to finish.
In their study, the authors collected data on university students at two-time points with a 10-week
interval to investigate relationships between proportions of future adverse events, levels of self-efficacy, and depressive symptoms. Results suggested that depressive symptoms predict subsequent proportions of negative prospects and levels of self-efficacy. Inversely, neither prospection nor self-efficacy predicted depression. Temporal correlations between prospection and self-efficacy were also not significant.
This study provides new insights into the relationship between negative biases in cognition and depressive symptoms. It highlights the need to intervene in early depressive symptoms before possible cognitive distortions in college students.
The work has merit for publication.
Author Response
Thank you for your kind comments. We have checked the manuscript again carefully and made some minor changes to improve it.
Reviewer 2 Report
The authors addressed the relationships between prospection, self-efficacy, and depression with cross-lagged analysis. This is an interesting study and can make practical and theoretical contributions. The manuscript is overall well written, but I have a few suggestions for publication.
The title can be rewritten to “The relationships between prospection, self-efficacy, and depression in college students with cross-lagged analysis.” I don’t think the authors need to highlight “temporal relationship”, “negative bias in prospection”, and “low self-efficacy.” More nondirectional title is more appealing. It’s my suggestion.
The introduction is concise and informative and presents key information to build the need and the purpose of the study. However, in the section “prospection and self-efficacy” the authors didn’t talk about gender effects. I recommend adding the information in the 1.3 section. In the section 1.4, the author didn’t talk about gender effects. I recommend adding the information in the section.
In line 117, delete ‘approach’
In lines 140, ‘sentence stems’ –>‘sentences’
In line 162 ‘specificity’à’specific’
In line 168, mediatoràmoderator
I can’t find why the authors design 10 weeks between T1 and T2. Presenting the rationales of that in the Introduction or Discussion section.
Overall In the discussion, theoretical contribution is well written with supporting previous studies and theories. But practical implication is lacking. I recommend the authors to strengthen the practical implication of the study throughout the Discussion.
